# Simultaneous Biological Pretreatment and Saccharification of Rice Straw by Ligninolytic Enzymes from *Panus neostrigosus* I9 and Commercial Cellulase

**DOI:** 10.3390/jof7100853

**Published:** 2021-10-12

**Authors:** Ariyah Terasawat, Sivawan Phoolphundh

**Affiliations:** Department of Microbiology, Faculty of Science, King Mongkut’s University of Technology Thonburi, 126 Pracha-Uthid Road, Bang Mod, Thungkru, Bangkok 10140, Thailand; Ariyah.tera@mail.kmutt.ac.th

**Keywords:** laccase, white-rot fungi, continuous fermentation, biological pretreatment

## Abstract

The utilization of rice straw for biofuel production is limited by its composition. The pretreatment process is required to improve the enzymatic accessibility of polysaccharides in the biomass prior to enzymatic saccharification. In this study, simultaneous biological pretreatment and saccharification (SPS) of rice straw starting from laccase production by *Panus neostrigosus* I9 was operated in a 2-L fermenter. It was found that fungal physiology was strongly influenced by the agitation, and that the highest laccase production was obtained at an agitation speed of 750 rpm (209.96 ± 0.34 U/L). The dilution rate of 0.05 h^−1^ was set in continuous fermentation which resulted in laccase activity of 678.49 ± 20.39 U/L, approximately three times higher than that in batch culture. Response surface methodology (RSM) was applied to achieve the condition for maximum percentage of delignification. The maximum percentage of delignification of 45.55% was accomplished after pretreatment of rice straw with laccase enzyme 39.40 U/g rice straw at 43.70 °C for 11.19 h. Reducing sugar of 3.85 ± 0.15 g/L was obtained from the digested rice straw in a SPS reactor, while non-pretreated rice straw gave only 1.13 ± 0.10 g/L within 12 h of incubation. The results indicated that simultaneous biological pretreatment and saccharification (SPS) of rice straw by laccase helped to improve the accessibility of cellulose by cellulolytic enzymes.

## 1. Introduction

Thailand has a variety of lignocellulosic biomasses, with rice straw being the most abundant agricultural waste. The utilization of rice straw for biofuel production is limited by its composition [1], as it consists of cross-linked polymers of cellulose, hemicellulose, and lignin. These structures provide impermeability and resistance to oxidative stress and microbial attack [2]. Therefore, structural modification and removal of lignin during the pretreatment process are required to improve the enzymatic accessibility of polysaccharides in the biomass prior to enzymatic saccharification and subsequent fermentation.

Acid and alkaline pretreatments effectively solubilize lignin and hydrolyze hemicellulose, but acid causes corrosion to the internal structures of the bioreactor [3] while alkali removes uronic acid substitutions on hemicellulose, reducing the accessibility of hemicellulose to hydrolytic enzymes [4]. The use of steam explosion for biomass pretreatment requires a high energy input and leads to the destruction of the xylan fraction, the formation of toxic and inhibitory phenolic compounds, and incomplete digestion of the lignin-carbohydrate matrix [5]. Biological pretreatment is a new type of pretreatment that uses enzymes to digest the complex structure of lignocellulose. Since highly specific enzymes are used in this process, it results in a high level of digestible solids, low sugar degradation, low toxicity, low energy consumption, fermentation compatibility, simplicity, and environmental friendliness. The ligninolytic enzymes were used for biological pretreatment. However, it is not easy to find the enzyme suitable for this process because slow degradation rates and possible consumption of the substrate by the organisms have been observed [3,6,7]. The optimized operating parameters can reduce the pretreatment time and increase the yield of monomeric sugar at minimal cost. Among these organisms, filamentous fungi and especially white rot basidiomycetes are the most efficient lignin degraders [8].

White rot fungi have been studied for their biodegradability for several decades [6,9]. They are called selective degraders because they attack the lignin component of wood, while cellulose and hemicellulose are less affected. It is reported that white rot fungi mostly colonize dead or living wood and can degrade lignin efficiently under natural conditions [10]. Lignin was removed by ligninolytic enzymes, such as laccase, manganese peroxidase (MnP), and lignin peroxidase (LiP), leaving the valuable cellulose intact. Furthermore, many reports on these fungi show a diversity of lignin degradation efficiency, enzyme patterns and substrates that promote lignin degradation [11,12].

Laccases (benzenediol oxygen oxidoreductase) are multicopper phenol oxidases that oxidize phenolic compounds to phenoxyl radicals and oxidize some nonphenolic compounds in the presence of a mediator, such as 2,2-azinobis (3-ethylbenzthiazoline-6-sulfonate) (ABTS) or 1-hydroxybenzotriazole [8]. The first laccase was isolated from plants, but also found in some fungi and bacteria [13]. The largest amounts of laccase are produced by white rot fungi [14]. In general, laccases are ligninolytic enzymes secreted into the surrounding medium by these fungi for various processes, such as lignin degradation, sporulation, pigment production, fruiting body formation, and plant pathogenesis [15]. It is believed that the ligninolytic enzymes of whit rot fungi are produced during secondary metabolism and normally under nitrogen deficiency [10]. The ligninolytic enzymes are usually obtained from cultured white rot fungi by solid-state fermentation, but the problem is the slow cultivation of white rot fungi on wood and their sensitivity to growth conditions [16,17,18].

Liquid or submerged fermentation has also been used favorably for enzyme production. Submerged fermentation (SMF) is the growth of microorganisms in a liquid medium associated with a high concentration of nutrients and oxygen (aerobic conditions). The morphology that develops in filamentous fungi in submerged culture influences both production and bioreactor control [19]. The two morphological differences during cultivation affect the mass transfer within the fermenter tank. First, the shape of the free mycelia increases the apparent viscosity, which can alter the rheological behavior of the non-Newtonian fluid (shear thinning or pseudoplasticity) [20]. This reduces mixing and air distribution, leading to limitations in solid–fluid mass transfer and oxygen transfer in the liquid medium. Second, the pellet shape restricts mass transfer within the particles. These limitations can lead to regions with different growth patterns and substrates. The outer zone is metabolically active, while zones of low viability exist inside the pellet. Therefore, both the rheology of the fermentation broth and the performance of the bioreactor have a significant impact on the morphological growth forms of the fungi [21]. Usually, stirred tanks (STR) provide efficient mixing of the fermentation broth and ensure adequate oxygen transport in the tanks. Although high agitation ensures good mixing, it can also generate high shear forces that can affect filamentous microorganisms, such as biomass concentration, morphological changes, cell wall breakage, variations in the rate of synthesis of products, and overall growth rate [19,22,23].

Results from our previous work showed that *Panus neostrigosus* I9, a fast-growing white rot fungus, could produce high laccase enzyme activity when cultured in V8 medium which has tomato juice as main composition [24]. However, large production of ligninolytic enzymes at low cost is required for its application [25]. Therefore, tomato juice can be another good choice to use as a medium for economic laccase production. In addition, laccase production by *P. neostrigosus* I9 was investigated as a function of physicochemical parameters under submerged cultivation [26]. This can be achieved by operating the reactor in a continuous mode. When laccase is secreted by the fungi, the cells grown in perfusion act as a renewable catalyst for continuous production as fresh medium washes out the secreted products into the environment. Therefore, the feasibility and advantages of continuous operation of STR were also investigated for laccase production.

Normally, the pretreatment process must be separated with the hydrolysis process, which consumes more time, energy, and cost. The ligninolytic enzyme of *P. neostrigosus* I9 could be mixed with the cellulase enzyme to create the single-step simultaneous biological pretreatment and saccharification process (SPS). Therefore, continuous laccase production is required for the continuous process. The aims of this research are (1) to achieve high production of laccase enzyme from *P. neostrigosus* I9 in continuous fermentation; and (2) to optimize the pretreatment conditions for the production of reducing sugars from rice straw by ligninolytic enzymes and cellulolytic enzymes using response surface methodology (RSM) to achieve high digestible solids content, low sugar degradation, low toxicity, and low energy consumption in a single-step SPS. In particular, SPS by ligninolytic enzymes is another option for a less chemical and environmentally friendly process to use biomass as feedstock for biofuel production.

## 2. Materials and Methods

### 2.1. Microorganism

The white rot fungus I9 was obtained from the Department of Microbiology at KMUTT and identified as *Panus neostrigosus* at Mycology Laboratory, National Center for Genetic Engineering and Biotechnology, Thailand. The fungal colonies of *P. neostrigosus* I9 from stock cultures were transferred to potato dextrose agar (PDA) plates (Himedia Inc.) and incubated at 30 °C for four days.

### 2.2. Cultivation of Panus neostrigosus I9 in the Stirred Tank for Laccase Production

#### 2.2.1. Inoculum Preparation

Tomato juice medium was prepared by diluted tomato juice (Doi Kham Food Product Co., Ltd., Chiang Rai, Thailand) with distilled water at a ratio of 1:3 *v/v* (initial total carbohydrate approximately 15 g/L) and adjusted to pH 5.0. The 250-mL flask containing 100 mL of tomato juice medium was sterilized at 121 °C, 103 kPa for 15 min. Two plugs of *P. neostrigosus* I9 (diameter 1 cm) cultured on PDA plates at 30 °C for four days were transferred to each flask. The culture was incubated at 28 °C with shaking at 150 rpm for four days. One hundred milliliters of cultured *P. neostrigosus* I9 which had a biomass dry weight of approximately 0.5 g/100 mL were used as inoculum size in the stirred tank reactor.

#### 2.2.2. Cultivation of *Panus* *neostrigosus* I9 in Batch Fermentation

Batch culture was carried out in a 2-L stirred tank reactor containing 1.35 L of tomato juice medium (dilution 1:3 *v*/*v*) pH 5.0 (after sterilization). Prior to use, the empty bioreactor was autoclaved at 121 °C and a vapor pressure of 103 kPa for 15 min to prevent the contamination that may occur from the material. The tank was sterilized again after adding the tomato juice medium. The prepared inoculum (100 mL) was transferred to the reactor and 50 mL of sterilized tomato juice medium was added to the tank to increase the working volume to 1.5 L. The system was operated at 28 °C with one vvm aeration and 750 rpm agitation for 120 h. Every 4 h, two milliliters of sample was collected in triplicate and centrifuged at 10,000 rpm for 10 min at 4 °C. The supernatant was immediately assayed for laccase activity, while the precipitate was used for biomass analysis. The remain supernatant was stored at −20 °C for total carbohydrate analysis.

Data collected during batch fermentation were used to create a model of microbial growth and a model of laccase formation.

#### 2.2.3. The Model of Microbial Growth


t = Time (hour)x = biomass concentration (g/L)$x_{m}$ = maximum biomass concentration (g/L)$x_{0}$ = Initial biomass concentration (g/L)$\mu_{m}\;$= maximum specipic growth rate (h^−1^)


The microbial growth model follows the model of Elibol & Mavituna [27] (Equation (1)). A plot of $\ln\frac{x_{t}}{(x_{m}-x_{t)}}$ versus time (t) will give a line of the slope and the vertical intercept equal to $\mu_{m}$ and $-(\ln\frac{x_{m}}{x_{0}}-1)$. Moreover, $\mu_{m}$ and $x_{0}\;$ can be determined using $x_{t}$ and $x_{m}$, respectively, and the fungal growth model can thus be constructed. The initial biomass concentration ($x_{0}$) plays a crucial role in increasing the total biomass production. The rate of biomass production is determined by the characteristics of fungal growth. The specific growth rate (μ) depends only on the fungal characteristics.
(1)$$\ln\frac{x_{t}}{(x_{m}-x_{t)}}=\mu_{m}t-(\ln\frac{x_{m}}{x_{0}}-1),$$

#### 2.2.4. Model of Laccase Formation


A = Coefficient for growth-associated production rate (U/g)β = Coefficient for non-growth-associated production rate (U/(g⋅hr))E = laccase activity (U/L)E_0_ = Initial laccase activity (g/L)S = Total carbohydrate (g/L), s_0_ initial substrate concentration (g/L)


The basis of the kinetics of product formation was calculated using the equation Luedeking-Piret (Equation (2)).
(2)$$\frac{dE}{dT}=\alpha \frac{dx}{dt}+\beta x$$

And Zhao et al. [28] evaluated equation for laccase formation in Equation (3)
(3)E=αA(t)+βB(t)A(t)=x0{eμmt1−(x0xm)(1−eμmt)−1}B(t)=xmμmln{1−x0xm(1−eμmt)}
(4)$$\frac{dE}{dT}=\beta x_{m}$$

A slope value from a plot of E − βB(t) against time A(t) gives α. A is a growth-associated constant (U/g) and β is a biomass-associated constant (U/(g⋅hr)) when dx/dt = 0 and x = $x_{m}$. β can be obtained by means of Equation (4). Therefore, β was calculated by dividing the productivity rate and the maximum biomass concentration in the stationary phase. $Y_{EG}$ represents the laccase yield of glucose (U/g) (Equation (5)), and $Y_{EG}$ is the biomass-specific yield of laccase (U/g) (Equation (6)).
(5)$$Y_{EG}=\frac{E_{f}-E_{0}}{S_{0}-S_{f}}$$
(6)$$Y_{EX}=\frac{E_{f}-E_{0}}{x_{0}-x_{f}}$$

#### 2.2.5. Cultivation of *Panus* *neostrigosus* I9 in Continuous Fermentation

Like batch culture, continuous fermentation was carried out in a 2-L stirred tank reactor containing 1.35 L of tomato juice medium (dilution 1:3 *v*/*v*) pH 5.0 (after sterilization) at 28 °C with aeration of 1 vvm and agitation at 750 rpm. Continuous fermentation started after 24 h batch cultivation and then operated continuously at a dilution rate of 0.05 h^−1^. The volume of the liquid fermenter was controlled with a peristaltic pump so that the feed rate of the medium exactly matched the efflux rate. Five milliliters of samples were collected in triplicate every 12 h and centrifuged at 10,000 rpm for 10 min at 4 °C. The supernatant was immediately analyzed for laccase activity, while the precipitate was used for biomass analysis. The residues were stored at −20 °C for determination of total carbohydrate.

Continuous laccase fermentation yielded a laccase activity of 678.49 U/L, with a residence time of 20 h. The fermentation broth was filtered with 0.45 μm of glass fiber filter paper and diluted two-fold with McIlvaine buffer pH 5.0 to serve as crude enzyme for biological pretreatment.

### 2.3. Optimization of Lignocellulosic Biomass Pre-Treatment Process

#### 2.3.1. Rice Straw Preparation

Sun-dried rice straw was obtained from a Thai rice farm and was chopped with a laboratory chopper to the length 10–15 cm. The gridded mixer was separeted using sieve at 40 mesh. The obtained rice straw powder (size approximately 0.4–0.5 mm) was dried at 105 °C until there was no change in weight and stored in an airtight desiccator.

#### 2.3.2. Optimization of Lignocellulosic Biomass Pre-Treatment Process

Two grams of dried, ground rice straw was pretreated with 100 mL of crude enzyme from *P. neostrigosus* I9 in a 250-mL Erlenmeyer flask. A Box–Behnken design with three levels (−1, 0, +1) and three factors (laccase concentration, (10, 25, 40 U/g rice straw); time for pretreatment (1, 6.5, 12 h); and temperature, (30, 45, 60 °C) (Table 1) were used to determine the optimum conditions of the pretreatment process for the delignification of lignocellulosic biomass using Design expert^®^ version 6.0 (Minneapolis, MN, USA). The coefficients of the response surface equation were determined using Design Expert^®^ Software.

#### 2.3.3. Pretreatment Method

Three pretreatment methods, i.e., steam explosion [29], acid treatment [30], and bio-pretreatment by crude enzyme from *P. neostrigosus* I9, was used to pretreat ground rice straw compared with non-pretreated rice straw.

##### Steam Explosion

Ten grams of ground rice straw was subjected to autocatalytic steam explosion at 180 °C and a vapor pressure of 900 kPa for 4 min, and the solid fraction of the pretreated slurry was collected by vacuum filtration. The collected solid fraction was then dried at 70 °C until constant weight. The dried material was then hydrolyzed with a commercial cellulolytic enzyme (Celluclast^®^ 1.5 L, produced by *Trichoderma reesei*).

##### Acid Pretreatment

Ten grams of ground rice straw was subjected to deacetylation with 0.1 mol/L H_2_SO_4_ at 121 °C 103 Pka for 1 h (solid:liquid ratio 1:6). Then, the pretreated slurry was adjusted to pH 5.0 with 10 M of NaOH. The solid fraction of the pre-treated slurry was filtered and dried at 70 °C until constant weight. The dried material was then hydrolyzed with a commercial cellulolytic enzyme (Celluclast^®^ 1.5 L, produced by *Trichoderma reesei*).

##### Biological Pretreatment

Two grams of dried milled rice straw were pretreated with 100 mL of crude *P. neostrigosus* I9 enzyme under optimal conditions (incubation at 43 °C and 40 U/g rice straw for 11.37 h) in a 250-mL Erlenmeyer flask. The pretreated rice straw was dried at 70 °C until constant weight. The percentage of delignification and structure (FTIR analysis) before and after pretreatment were analyzed. The dried material was then hydrolyzed with a commercial cellulolytic enzyme (Celluclast^®^ 1.5 L, produced by *Trichoderma reesei*).

##### Enzymatic Hydrolysis

Enzymatic hydrolysis was carried out using commercial cellulolytic enzymes (Celluclast^®^ 1.5 L, produced by *Trichoderma reesei*). Two grams of unpretreated or pretreated rice straw was placed in a 250-mL Erlenmeyer flask containing 100 mL of phosphate buffer pH 6.0. The cellulolytic enzymes were added to the flask at a dose of 20 FPU/g rice straw after the pretreatment process. The reaction mixture was incubated in a rotary shaker at 150 rpm and 50 °C for 48 h. One milliliter sample was taken every 6 h and centrifuged at 10,000 rpm for 5 min. The supernatants were assayed for reducing sugars, as well as laccase and cellulase activity.

### 2.4. Simultaneous Pretreatment and Saccharification of Rice Straw

Milled rice straw was hydrolyzed simultaneously by mixing crude enzyme from *P. neostrigosus* I9 under optimization conditions and Celluclast^®^ (20 FPU/g rice straw) in a 2.0-L reactor tank at a working volume of 1.5 L and a rotation speed of 150 rpm. Non-pretreated rice straw was hydrolyzed with only 20 FPU/g rice straw of Celluclast^®^ in a 2.0-L reactor tank at a working volume 1.5 L and a rotation speed of 150 rpm. Samples (1 mL) were taken from each tank every 6 h and centrifuged at 10,000 rpm for 5 min. The supernatant was concentrated by tangential crossflow 10 kDA cut-off filtration (Pall corp.) and analyzed for reducing sugars and laccase activity.

### 2.5. Analysis Methods

#### 2.5.1. Biomass Composition of Rice Straw

The content of cellulose, hemicellulose, and lignin was analyzed according to the analytical procedure [31] in Standard Biomass Analytical Methods provided by National Renewable Energy Laboratory (NREL).

#### 2.5.2. Percentage of Delignification

The kappa number test was used to estimate the lignin content by measuring the oxidant demand of the pulp. Kappa numbers were performed on air-dried rice straw samples, which were dried on a heated balance to obtain their oven-dried weight. Rice straw was then treated with potassium permanganate (KMnO_4_) according to TAPPI Classical Method T 236 cm-85 “Kappa number of pulp” [32]. The kappa number was converted to Klason lignin using Equation (7). Finally, the percentage of delignification was calculated using Equation (8).
(7)Kl=K×0.13.
(8)$$\%delignification=\frac{Kl_{i}-Kl_{f}}{Kl_{i}}\times 100$$K = kappa numberKl = Kalson lignin%delignification = Percentage of delignification

#### 2.5.3. Enzyme Assay

Laccase activity was determined colorimetrically by modification of Rodriguez et al. [14]. The reaction mixture contained 1000 µL of crude supernatant, 100 µL 2 mM of ABTS in a 25-mM sodium succinate buffer pH 5 and 900 µL of deionized water. The oxidation of 2,2″-azino-bis-3-ethylbenzthiozoline-6-sulfonic acid (ABTS) was monitored by determining the increase in absorbance at 436 nm after one minute incubation at 30 °C. Extinction coefficient (ε) of laccase at 436 nm = 29,300 M^−1^cm^−1^. One unit of enzyme activity (U) is defined as the amount of enzyme that produces 1 µmol of product per minute under the assay conditions.

Cellulase activity was determined using the filter paper activity (FPA) assay [33]. The crude enzyme (100 µL) was incubated with 400 µL of 50-mM sodium citrate buffer (pH 5.0) containing filter paper Whatman No. 1 (5 mg/mL). The reaction mixture was incubated at 50 °C for 30 min and the reaction was stopped by adding 500 µL of 3,5-dinitrosalicylic acid (DNS) reagent. The solution was placed in a water bath at 95 °C for 5 min, then 2 mL of distilled water was added, and the absorbance was measured using a spectrophotometer at 540 nm. The absorbance was converted to concentration using the standard glucose curve. One unit of enzyme activity (U) is defined as the amount of enzyme that produces 1 µmol of reducing sugar per minute under the assay conditions.

#### 2.5.4. Total Carbohydrate Assay by Phenol–Sulfuric Method

According to the phenol–sulfuric method [34], the total carbohydrate was determined using the 5% phenol solution and D-glucose as standard. One milliliter of the sample was mixed with 1 mL of 5% phenol solution and 5 mL of 98% sulfuric acid was added, boiled for 5 min, and cooled at room temperature. The solution was measured at 480 nm using a spectrophotometer. The absorbance was converted to concentration using the standard glucose curve (0–100 mg/L).

#### 2.5.5. Reducing Sugar Assay by DNS Method

Total reducing sugar was determined by the 3,5-dinitrosalicylic acid (DNS) assay [35] using D-glucose as standard. Next, 500 µL of the sample was mixed with 500 µL of DNS. The solution was placed in a water bath at 95 °C for 5 min. Then, 2 mL of distilled water was added, and the solution was measured using a spectrophotometer at 540 nm. The absorbance was converted to concentration using the standard glucose curve. The yield of reducing sugar was calculated using the following Equation (9).
(9)$$\mathrm{Reducing}\;\mathrm{sugar}\;\mathrm{yield}\;\left(\%\right)=\frac{Reducing\;sugar\;\left(\frac{g}{L}\right)\times 0.9\times 100}{thgiew\;of\;rice\;straw\;\left(\frac{g}{L}\right)}$$

#### 2.5.6. FTIR Spectrum Analysis

Fourier Transform Infrared Spectroscopy of the untreated and pretreated rice straw was recorded using FTIR Thermo Nikolet 6700 (Waltham, MA, USA). Samples were used in the form of KBr discs prepared by grinding 1 mg of sample/100 mg of pre-dried KBr. The spectra were recorded in the range of 400–4000 cm^−1^ [7]. The obtained FTIR spectra were compared in terms of shift of peaks for different functional groups associated with different components of lignocellulosic biomass.

### 2.6. Statistical Analysis

All experiments were performed in triplicate and the values are the average of three values. The reducing sugar production during biological pretreatment and hydrolysis was analyzed using analysis of variance (ANOVA) (with a confidence level of 95%).

## 3. Results

### 3.1. Determination of Kinetic Models of P. neostrigosus I9

Batch fermentation of *P. neostrigosus* I9 was carried out in 2.0-L STR with working a volume of 1.5 L at 750 rpm, pH 5.0, and 28 °C. The dry weight of the fungus and laccase activity are shown in Figure 1.

#### 3.1.1. The Model of Fungal Growth Kinetics

The increase in fungal dry weight was consistent with the increase in laccase activity (Figure 1). Total carbohydrate was depleted after 60 h of cultivation, while laccase activity was detected from 20 h of cultivation and maintained high activity until 80 h of cultivation (Figure 1). The maximum specific growth rate (µ_max_) of *P. neostrigosus* I9 was determined by batch fermentation.

From the results, the specific growth rate (µ) could be determined using Equation (1). The dry weight of *P. neostrigosus* I9 in batch fermentation was plotted against cultivation time, and the maximum specific growth rate (µ_max_) was determined by the slope of the tangent line during a rapid increase in dry weight (Figure 2). At the beginning, the biomass entered a lag phase followed by an exponential growth phase in the growth curve. Biomass concentration and laccase activity increased simultaneously as the fungus entered the exponential phase. Assuming that x_max_ = 6.70 g/L, a plot of Equation (1) corresponds to x_0_ = 0.79 g/L. The maximum specific growth rate (µmax) of *P. neostrigosus* I9 was 0.075 h^−1^, and this value was used to determine the dilution rate (D) in the continuous culture. In this work, the dilution rate of the continuous fermentation was set lower than the maximum specific growth rate to avoid washout.

#### 3.1.2. The Kinetic Model of Laccase Formation

The basis of the kinetics of product formation is expressed by the Luedeking–Piret equation. α and β are constants related to culture conditions. Therefore, from the results in Figure 1 (at 56 to 76 h of cultivation) (Equation (4)), β = 0.497 U/(g h) was calculated. Finally, a plot of E − β B(t) against time A(t) (Figure 3) gives a line with slope α = 16.163 U/g (Equation (3)).

The results presented in Figure 3 showed that the α value is 16.163 U/g cell compared to β = 0.497 U/(g h), which means that the laccase production associated with the growth of *P. neostrigosus* I9 and was greater during cell growth in the exponential phase than in the stationary phase. Therefore, if the exponential phase can be prolonged, it is possible that laccase production can also be increased.

### 3.2. Cultivation of P. neostrigosus I9 in Continuous Culture for Laccase Production

By results from batch culture, the maximum dilution rate of continuous culture was 0.075 h^−1^. Therefore, continuous fermentation of *P. neostrigosus* I9 was carried out in a 1.5-L fermenter broth using a tomato juice medium (dilution 1:3 *v*/*v*) from batch fermentation. *P. neostrigosus* I9 was batch cultured for 24 h and then switched to continuous cultivation at a dilution rate of 0.05 h^−1^. The tomato juice medium was fed to the fermenter using a peristaltic pump. The retention time (RT) of the cultivation at a dilution rate of 0.05 h^−1^ was 20 h. Samples were taken every 12 h until steady state (about 3 RT or 60 h of cultivation).

To achieve high productivity of laccase enzyme, continuous fermentation was performed. For chemostat culture, the fermenter was operated batchwise until a moderate cell concentration was reached and then operated continuously. Dry weight, laccase activity, and total carbohydrate during fermentation are shown in Figure 4. At a dilution rate of 0.05 h^−1^, the dry weight at the switchover point (24 h) of batch fermentation was 6.94 g/L. The dry weight increased while the total carbohydrate decreased with cultivation time. (Figure 4). The steady state of the culture was reached between 36 and 84 h after cultivation. At the midpoint of steady state, the concentrations of biomass and residual total carbohydrate were 6.04 g/L and 5.38 g/L, respectively (60 h) (Figure 4). At steady state, stabilization of dry weight was observed from 6.04 to 7.12 g/L (Figure 4). However, cell accumulation was observed after 96 h because *P. neostrigosus* I9 could not be removed from the fermenter due to the low efflux rate. The highest laccase activity of 678.49 ± 20.39 U/L was reached after 60 h and remained constant until 108 h after cultivation.

### 3.3. Effect of Biological Pretreatment Conditions on Delignification

Rice straw is a lignocellulosic material that could be a promising resource for biorefineries because it is abundant, inexpensive, and not a food. The composition of rice straw was analyzed and shown in Table 2. It can be seen that hemicellulose is the major component (48.667 ± 1.790%) in rice straw, followed by cellulose (30.671 ± 1.393%) and lignin (15.327 ± 1.175%) as minor components. The composition of the raw rice straw indicates a potentially valuable source of fermentable sugar.

The aim of pretreatment is to remove lignin from the lignocellulosic biomass and to improve the accessibility of cellulose by cellulolytic enzymes. The pretreatment process should be able to increase digestibility to produce reducing sugars by cellulolytic enzymes and limit the formation of fermentation inhibitors. From our previous studies, the optimal conditions for laccase from *P. neostrigosus* I9 were 60 °C and pH 3.0 (Appendix A). However, this laccase could not be active for long at a pH of 3.0 (laccase activity at 60 °C and a pH of 3.0 decreased by 50% within 40 min) 0 (Appendix A). Therefore, the pH kept at pH 5.0, laccase concentration (10–40 U/g substrate), pretreatment time (1–12 h), and temperature (30–60 °C) were chosen as experimental design factors to optimize biological pretreatment by ligninolytic enzymes. The effects of the levels of the selected variables for the Box–Behnken design and summaries on the percentage of delignification of pretreated rice straw are shown along with the predicted values in Table 3.

The adequacy of the model was assessed with ANOVA at a moderate coefficient of determination (R^2^ = 0.9929), which means that 99% of the variability of the response could be predicted by the model, only 1.0% of all variations for the response could not be explained by the model and expresses a good fit. The model F-value of 108.66 and a low probability value (*p* < 0.0001) showed that the model terms were significant. Adequate precision represents the signal-to-noise (S/N) ratio, and values more than 4.0 indicated that the model precision is adequate. Adequate precision ratio of 37.293 indicated an adequate signal.

The overall effect of the three factors on delignification analyzed by a joint test (Table 4) showed that temperature, incubation time, and laccase concentration had statistically significant effects on delignification (*p*-values < 0.0001). The interactions between AB, A, C and BC have significant effects on delignification. The three-dimensional response surfaces showing the relationship between factors on lignin content in rice straw are shown in Figure 5.

Figure 5a showed the positive effect of time on percent delignification. The percentage of delignification increased with increasing temperature and time. However, at temperatures higher than 48 °C, delignification decreased with increasing time. Figure 5b showed the positive effect of low temperature on percentage of delignification. However, delignification decreased with increasing temperature above 45 °C. Finally, Figure 5c showed the positive effect of high laccase concentration and long period on percentage of delignification. The percentage of delignification decreased with decreasing laccase concentration under 28 U/g of rice straw.

The response surface regression procedure in the software Design Expert was used to fit the following second-order polynomial Equation (10). A statistically significant model only with significant terms can be written as follows:%delignification = −54.31 + (4.877A) + (6.86B) − (3.074C) − (0.111AB) + (0.043AC) + (0.06BC) − (0.06A^2^) − (0.182B^2^) + (0.02C^2^)(10)

According to Equation (10), the factors predicted for optimum delignification (50.28%) were (temperature) 47 °C, (incubation time) 11.27 h, and (laccase concentration) 38.00 U/g dry weight. The observed delignification value using the predicted factor was 45.55%, which is close to the predicted value (temperature 45 °C, incubation time 12 h and laccase concentration 40 U/g dry weight)

### 3.4. Reducing Sugar Production by Using Optimized Biological Pretreatment and Saccharification Processes of Rice Straw

Reducing sugar concentration and productivity of reducing sugars (yield) from different pretreatment methods are shown in Table 5. Steam explosion pretreatment showed the highest digestibility (reducing sugar 4.75 ± 0.15 g/L within 24 h), while enzyme-pretreated rice straw and non-pretreated rice straw gave 3.06 ± 0.14 g/L and 1.57 ± 0.32 g/L within 36 h and 12 h, respectively. The lowest content of reducing sugars (0.27 ± 0.01 g/L) was recorded when the acid-pretreated rice straw was digested by cellulolytic enzymes. The steam explosion method still gave the highest productivity (23.74%) while the biological pretreatment, non-pretreatment, and acid pretreatment gave 15.31%, 7.58%, and 1.36%, respectively. Although biological pretreatment was not the best pretreatment method, it can increase digestibility by cellulolytic enzymes. Therefore, biological pretreatment by ligninolytic enzymes is another option for a low-energy, less chemical, and environmentally friendly process to use biomass as feedstock for biofuel production.

### 3.5. Effect of Biological Pretreatment on Rice Straw Morphology

The morphology of the rice straw in the non-pretreated and the biologically pretreated samples was analyzed using FT-IR spectroscopy (Figure 6). The shape of the 3700–3000 cm^−1^ caused broad band changes in the pretreated rice straw, indicating a change from intermolecular to intramolecular OH bonds. In the fingerprint region, FT-IR showed the same pattern in untreated and pretreated rice straw. This means that both still had the same chemical component and the other chemical structure was not changed. The -OH stretching in lignin at 3421 cm^−1^ was decreased as indicated. The intensity of the lignin bands at 1593, 1504, and 1234 cm^−1^ decreased slightly, and the intensity of the band at 1647 cm^−1^, which is conjugated with carbonyl groups and is mainly from lignin, was also lower. Whereas, the peaks at 1428 and 1458 cm^−1^ indicated increasing deformation of lignin and carbohydrates [36]. The bands at 1369, 1158, 875, and 930 cm^−1^ assigned to the glycosidic linkage of hemicellulose and cellulose were altered. In short, the lignin structure in rice straw was attacked by the enzyme, and the structure of cellulose and hemicellulose in rice straw was also changed by the enzyme during pretreatment.

### 3.6. Reducing Sugar Production from Simultaneous Biological Pretreatment and Saccharification

The conditions in the reactor for simultaneous pretreatment and saccharification (SPS) were obtained from the results of RSM of pretreatment. The rice straw was pretreated and hydrolyzed simultaneously by laccase and cellulolytic enzymes. The characteristics of the fermenter broth (laccase enzyme production) and hydrolysis reactor broth are shown in Table 6.

The results showed that rice straw was digested as well as the reducing sugar 3.85 ± 0.15 g/L could be detected in the SPS reactor, while non-pretreated rice straw gave only 1.13 ± 0.10 g/L within 12 h of incubation (Figure 7). This indicated that the enzymatic hydrolysis yield of rice straw was affected by the enzymatic pretreatment process.

The concentrated sugars by tangential crossflow 10 kDA cut-off filtration (Pall corp.) gave a reducing sugar of 22.27 g/L and 0.72 g/L in the filtrate and the retentate, respectively, which was about six times more concentrated than before. On the other hand, laccase activity of 283.96 U/L detected in the retentate (Table 6) showed that crossflow filtration can recover about 90% of the laccase activity. It is possible that the laccase enzyme can be reused for another pretreatment.

## 4. Discussion

### 4.1. Cultivation of P. neostrigosus I9 in Continuous Culture for Laccase Production

In the previous study, tomato juice medium was successfully used for laccase production by *P. neostrigosus* I9 in shake flasks (412.54 U/L) [24] (Table 7). Therefore, the tomato juice medium was subsequently tested for laccase production in 2-litre fermenters. It was found that the fungal physiology was strongly influenced by the agitation conditions used, resulting in relatively low laccase production. The laccase activity of *P. neostriogsus* I9 in the shake flask (412.54 U/L) was higher than that in the fermenter tank (209.96 ± 0.03 U/L) because mechanical stress in the fermenter tank may affect enzyme production.

The values of α and β depend on the fermentation conditions. The results showed an α- and β-value of 16.163 U/g cell and 0.497 U/(g h), respectively. α is a growth-associated constant (U/g), which can be considered as the laccase production of the young-age mushrooms, and β is a biomass-associated constant (U/(g h)), which means the laccase production coefficient of the older-age mushrooms [28]. This means that the laccase production of *P. neostrigosus* I9 during cell growth was higher in the exponential phase than in the stationary phase.

*P. neostrigosus* I9 is a fast-growing white rot fungus when cultivated in submerged culture, especially in tomato juice medium. When the agitation rate is above the optimal rate, and fungal growth is inhibited due to insufficient mixing and hydrodynamic shear stress [37,38]. Although *P. neostrigosus* I9 could be tolerated at a high agitation speed when compared to the other white rot fungi (Table 8), the positive correlation turns into a negative one when the rotation rate exceeds 750 rpm, indicating that increasing the agitation speed does not promote enzyme formation [26] (Appendix A). This is because fungal morphology in fermenters is also important for energy and oxygen transport [19,39,40,41]. Furthermore, in this study, laccase activity reached its maximum (209.96 ± 0.03 U/mL) at a stirring speed of 750 rpm.

The laccase activity of *P. neostrigosus* I9 cultured in continuous condition in steady state was 678.49 ± 20.39 U/mL and the productivity was 93.82 U/hour. These results were consistent with the predictions of the kinetic model of laccase formation; when the exponential phase was prolonged, laccase production was also increased.

Although the medium flow was continuously fed the reactor, the carbohydrate concentration was not constant because the growth of the fungal walls in the fermenter tank did not come out with the feed out but accumulated in the tank and continued to consume substrate. Numerous mycelial clumps were formed everywhere in the fermenter, such as pH and DO probes, and the reactor vessel wall. Therefore, the carbohydrate content decreased slightly after the feeding flow. This problem resulted in uncontrollable growth in continuous culture after 84 h of cultivation. Moreover, this problem should be solved in the future experiments.

The results of optimization could be assessed by means of the kinetic parameters, aside from assessing maximal laccase activity and fermentation duration. These parameters are conducive to a better understanding of the fermentation process and can serve as a reference for enhancement of enzyme production. Therefore, prolonging the exponential phase of *P. neostrigosus* I9 may increase laccase production. *P. neostrigosus* I9 gave quite low laccase activity compared to other studies, but white rot fungi could be stimulated to produce laccase by inducers, such as Cu^+2^ and veratryl alcohol, so this method can increase laccase production by adding it to the fermenter (Table 8).

### 4.2. Reducing Sugar Production from Simultaneous Biological Pretreatment and Saccharification

The overall effect of the three factors on delignification analyzed by a joint test (Table 5) showed that temperature, incubation time, and laccase concentration had statistically significant effects on delignification (*p*-values < 0.05). The interactions between AB, AC, and BC have significant effects on delignification. The positive effect of low temperature and time gave positive effects on percentage delignification. However, at temperatures above 54 °C, delignification also decreased with increasing time. Zang et al. [53] and Quaratino et al. [54] also reported the maximum laccase activity of *Panus rudis* at 60 °C and *Panus triginus* at 55 °C, respectively. The same results showed that the laccase enzyme of *P. neostrigosus* I9 has a higher activity at 60 °C than at 30 °C, although this high activity can only be maintained for a short period of time. On the other hand, the activity of laccase of *P. neostrigosus* I9 was better maintained at 30 °C than at 60 °C (Appendix A). The percentage of delignification decreased with decreasing laccase concentration under 28 U/g of rice straw. This could be due to the fact that laccase enzyme denatured during the pretreatment process. Up to a certain point, the laccase concentration was not sufficient to enter the process, resulting in lower percentage delignification at low laccase concentration. The predicted delignification response was 50.28%, while the observed delignification values using the predicted conditions were 45.55%. The FTIR study showed that the lignin structure, cellulose, and hemicellulose in rice straw changed during pretreatment by the enzyme.

In the hydrolysis phase, the rice straw pretreated with the biological pretreatment method was digested better by cellulolytic enzymes than non-pretreated rice straw. All these results indicated that pretreatment with ligninolytic enzymes is an effective method for pretreating biomass as feedstock for biofuel production, when compared with other pretreatment processes. The lowest content of reducing sugars (less than 0.5 g/L) was found when the rice straw pretreated with acid was digested by cellulolytic enzymes. This could be due to the effect of sulfuric acid which produces inhibitors of cellulolytic enzymes, such as furfural (2-furaldehyde), 5-hydroxymethylfurfural (5-HMF), 4-hydroxybenzaldehyde, and syringaldehyde [55]. Recently, Hu et al. [56] reported that pseudolignin is produced during the repolymerization of lignin because the digestion by acid treatment was incomplete, and this compound can inhibit cellulolytic enzymes. Therefore, these substances must be removed before hydrolysis.

The reducing sugar yields reported by other researchers during enzymatic hydrolysis of various lignocellulosic biomasses and pretreatment processes are summarized in Table 9. Biological pretreatment yields quite a small reducing sugar compared to the chemical process. However, compared to biological pretreatments and some chemical pretreatments, our techniques achieve quite good results. The yield of reducing sugar depends on the extent of delignification of the lignocellulosic material as the physical protective layer of cellulose is removed, resulting in improved digestibility of cellulose [57,58]. The results showed that rice straw is a potential substrate that yields the highest amount of sugar under optimized conditions.

Compared to other biological pretreatment methods (Table 9), a distinction must be made between pretreatment and saccharification. In other techniques, the fungi are directly inoculated into the lignocellulosic material, which consumes more time, energy, and cost. The techniques of SPS could combine pretreatment and saccharification in a single step by mixing the crude enzyme with commercial cellulase. Therefore, this technique saves time and energy during the process. Moreover, SPS by ligninolytic enzymes is another option for a less chemical and environmentally friendly process to use biomass as feedstock for biofuel production.

## 5. Conclusions

The kinetic model showed that *P. neostrigosus* I9 can produce more laccase in the extensive log phase. Therefore, a continuous process could allow high laccase production. This study highlights the potential use of rice straw, the most abundant agricultural waste, as feedstock for biofuel production. It also confirms the validity of RSM. Pretreatment of rice straw with the ligninolytic enzyme of *P. neostrigosus* I9 under conditions obtained from RSM enhanced a cellulose-rich substrate in an environmentally friendly manner, compared to non-pretreatment and other pretreatment methods. An enzyme mixture of laccase and cellulolytic enzymes could produce reducing sugars from rice straw with high digestible solids, low sugar degradation, low toxicity, and low energy consumption in a single step of the simultaneous pretreatment and saccharification process (SPS). High reducing sugar concentration, which was about six times more concentrated, was achieved by tangential crossflow 10 kDA cut-off filtration (Pall Corp.). Besides, crossflow filtration recovered about 90% of the laccase activity in the retentate indicated that the use of laccase enzyme in another time can be possible. Therefore, SPS by ligninolytic enzymes is another option for a less chemical and environmentally friendly process to use biomass as feedstock for biofuel production.

## Figures and Tables

**Figure 1 jof-07-00853-f001:**
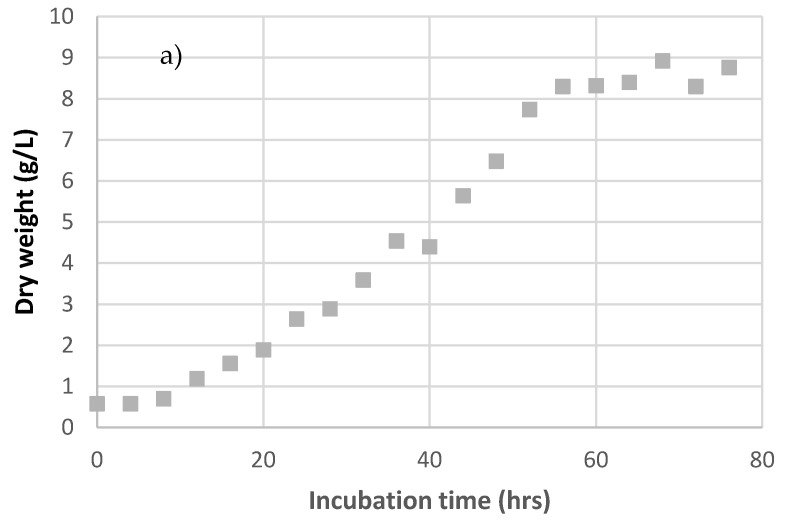
Time course study of dry weight (**a**) and laccase activity (**b**) during batch cultivation of *P. neostrigosus* I9 in 1.5 L tomato juice medium (diluted 1:3 *v*/*v*) at 28 °C, 750 rpm.

**Figure 2 jof-07-00853-f002:**
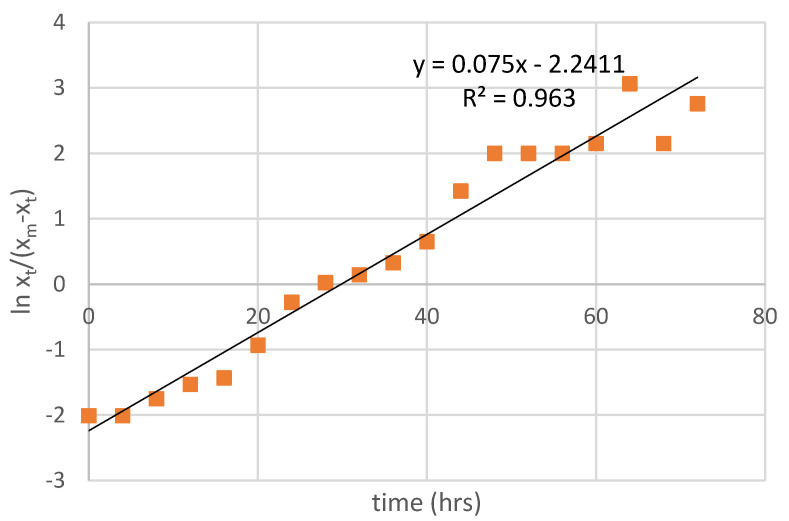
Determination of the maximum specific growth rate (µ_max_) of *P. neostrigosus* I9 cultured in 1.5 L tomato juice medium (diluted 1:3 *v*/*v*) in batch culture at 28 °C and 750 rpm.

**Figure 3 jof-07-00853-f003:**
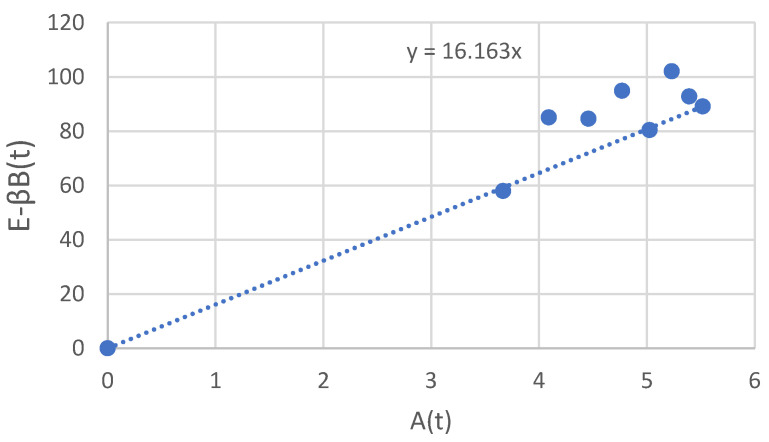
Determination of α. The slope of the line was α (U/g), the growth-associated constant for laccase formation. β, the biomass-associated constant for laccase formation, was equal to 0.497 U/g/d).

**Figure 4 jof-07-00853-f004:**
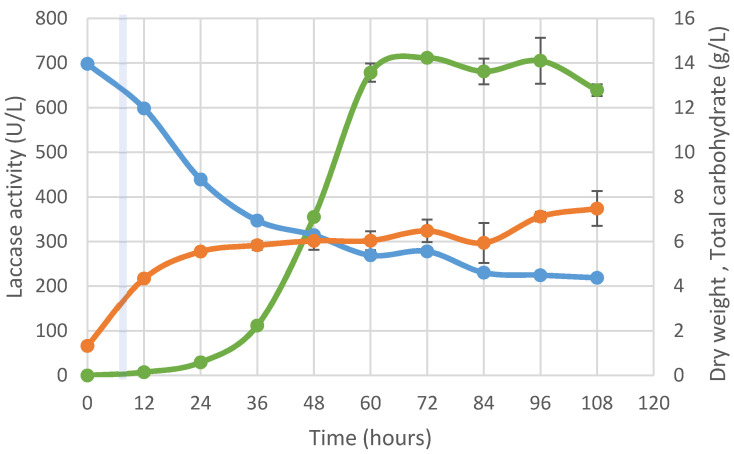
Time course of dry weight (orange), total carbohydrate (blue) and laccase activity (green) during continuous cultivation of *P. neostrigosus* I9 in tomato juice medium at a dilution rate of 0.05 h^−1^.

**Figure 5 jof-07-00853-f005:**
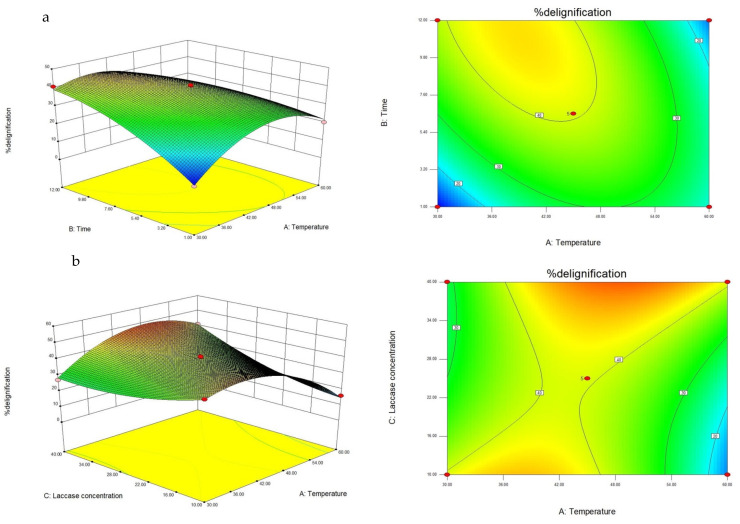
Contour and surface plots of the interaction between time and temperature (**a**), temperature and laccase concentration (**b**) and time and laccase concentration (**c**) on the percentage of delignification.

**Figure 6 jof-07-00853-f006:**
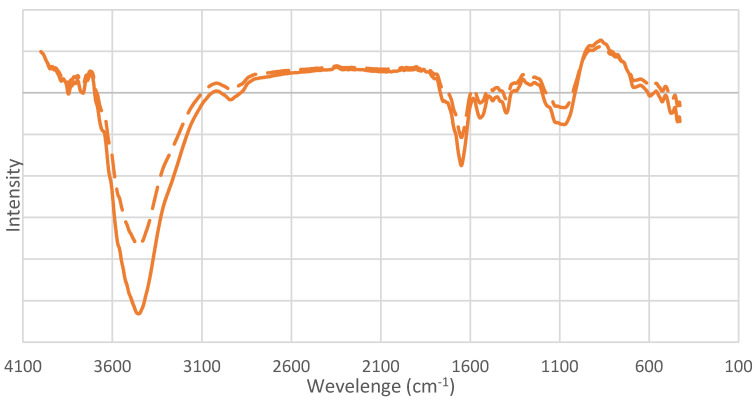
FT-IR spectra of non-pretreated (solid line) and biologically pretreated rice straw (dash line).

**Figure 7 jof-07-00853-f007:**
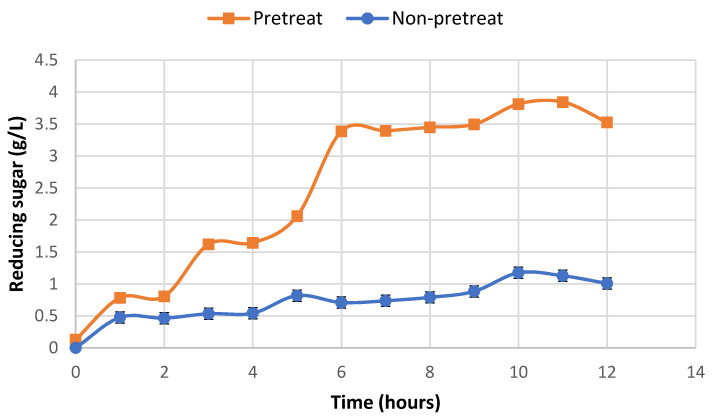
Time course of reducing sugar concentration during hydrolysis with simultaneous biological pretreatment and saccharification of rice straw and untreated rice straw.

**Table 1 jof-07-00853-t001:** Factors level for design a Box–Behnken experiment of delignification of rice straw.

Factor	Low	Medium	High
	−1	0	1
Temperature	30	45	60
time	1	6.5	12
Laccase concentration	10	25	40

**Table 2 jof-07-00853-t002:** Biomass composition of rice straw.

%	Acid Soluble Lignin	Acid Insoluble Lignin	Total Lignin	Cellulose	Hemi-Cellulose	Ash	%Recovery
Rice straw	4.88 ± 1.26	10.45 ± 0.22	15.33 ± 1.18	30.67 ± 1.39	48.67 ± 1.79	2.99 ± 0.79	97.66 ± 1.62

**Table 3 jof-07-00853-t003:** Factors of the RSM experimental design. Percentage of delignification as a function of rice straw factors.

Run Order	Factors			Response Y1 %Delignification	
	A (Temp, °C)	B (Time, Hour)	C (Laccase Conc., U/g Sub)	Observed Values	Predicted Values
1	30	1.00	25	9.567	10.748
2	60	1.00	25	20.278	23.978
3	30	12.00	25	40.666	40.052
4	60	12.00	25	14.472	16.652
5	30	6.50	10	40.801	41.315
6	60	6.50	10	16.002	16.881
7	30	6.50	40	27.297	29.496
8	60	6.50	40	40.902	43.761
9	45	1.00	10	35.861	36.548
10	45	12.00	10	34.419	37.637
11	45	1.00	40	34.406	34.178
12	45	12.00	40	52.699	55.067
13	45	6.50	25	40.078	41.863
14	45	6.50	25	39.923	41.863
15	45	6.50	25	40.516	41.863
16	45	6.50	25	41.112	41.863
17	45	6.50	25	40.111	41.863

**Table 4 jof-07-00853-t004:** Experimental matrix and results of delignification of rice straw.

Source	Sum of Squares	df	Mean Square	F Value	*p*-Value Prob > F
Model	2209.22	9	245.47	108.66	<0.0001
A-Temp	88.95	1	88.95	39.38	0.0004
B-Time	222.02	1	222.02	98.28	<0.0001
C-laccase conc.	99.56	1	99.56	44.07	0.0003
AB	340.49	1	340.49	150.72	<0.0001
AC	368.72	1	368.72	163.21	<0.0001
BC	97.37	1	97.37	43.1	0.0003
A2	778.69	1	778.69	344.69	<0.0001
B2	127.52	1	127.52	56.45	0.0001
C2	85.32	1	85.32	37.77	0.0005
Residual	15.81	7	2.26		
*Lack of Fit*	14.89	3	4.96	21.56	0.0062
*Pure Error*	0.92	4	0.23		
Cor Total	2225.03	16			
Std. Dev.	1.5	R-Squared	0.9929		
Mean	33.48	Adjusted R-Squared	0.9838		
C.V %	4.49	Predicted R-Squared	0.8923		
PRESS	239.72	Adequate Precision	37.293		

**Table 5 jof-07-00853-t005:** Results of steam explosion, acid treatment, and biological treatment.

Rice Straw	Reducing Sugar (g/L)	Time (Hour)	Yield %, Reducing Sugar/g Substrate
Steam explosion	4.75 ± 0.15	24	23.74 ^a^
Laccase pretreatment	3.06 ± 0.14	36	15.31 ^b^
Acid treatment	0.27 ± 0.01	36	1.36 ^d^
Non-pretreatment	1.57 ± 0.32	12	7.58 ^c^

Note: Means of yield (%, reducing sugar/g substrate) with difference superscript letter indicated significant difference (*p* ≤ 0.05) according to Duncan’s multiple range test.

**Table 6 jof-07-00853-t006:** Reducing sugar production from non-pretreated and pretreated rice straw.

		Laccase Activity (U/L)		Volume (L)		Rice Straw (g/L)	Cellulolytic Enzymes Conc. (FPU/g)	Reducing Sugar (g/L)			Theoretical Sugar	Saccharification
		Initial	Final	Initial	Final			Initial	Final	Production	g/L	%
Fermenter broth		678.49	N/A	1.5	1.5	N/A	N/A	5.02	N/A	N/A	N/A	N/A
SPS rice straw broth		339.24	317.18	1.5	1.5	8.72	20	2.47	6.32	3.85	6.92	55.49
Non-pretreat rice straw		0	0	1.5	1.5	8.72	20	0	1.13	1.13	6.92	16.30
Crossflow Concentrator	filtrate	317.18	0	1.5	0.25	N/A	N/A	3.84	22.27	N/A	N/A	N/A
	Retentate	317.18	283.96	1.5	1.25	N/A	N/A	3.84	0.72	N/A	N/A	N/A

Note: N/A: not available.

**Table 7 jof-07-00853-t007:** The effects of mode of operation on the kinetic characterization of fermentation.

Operation	Agitation (rpm)	Time (Hours)	Dry Weight (g/L)	μ_max_	Laccase Activity (U/L)	Laccase Absolute Activity (U)	Productivity U/hr	Y_EG_	Y_EX_	Reference
shake flasks	150	144	N/A	N/A	412.54	N/A	N/A	N/A	N/A	[24]
batch	750	72	7.60 ± 0.11	0.075	209.96 ± 0.03	314.94	4.37	16.88	32.56	This study
continuous	750	60 Std	5.83 ± 0.20	0.05	678.49 ± 20.39	5629.23	93.82	N/A	N/A	

Note: N/A: not available.

**Table 8 jof-07-00853-t008:** Maximum laccase activities obtained from different filamentous fungi at bioreactor scale.

Fungus	Type of Reactor	Type of Cultivation	Inducer	Max. Laccase Activity (U/L)	Reference
*Pycnoporus cinnabarinus*	10-L packed-bed	SmF, immobilised on nylon cubes	10 mM VA	280	[42]
*Trametes pubescens*	20-L STR (150 rpm)	SmF, free cells	2 mM Cu^+2^	61,900	[43]
*Neurospora crassa*	Capillary membrane	SmF, immobilised on membrane supports	1 μM cycloheximide	10,000	[44]
*Phanerochaete flavido-alba*	Bioflo III (975 mL ^a^; 70 rpm)	SmF, free cells	oil mill wastewaters	72	[45]
*Trametes hirsuta*	1-L fixed-bed	SmF (immobilised on stainless steel sponges)	Cu^+2^	2206	[46]
*Trametes versicolor*	STR (1 L)	SmF	30 μM xyldine	11,403	[47]
*T. versicolor*	2-L STR (1.5 L)	SmF free cells	N/A	5.3	[48]
*T. versicolor*	5-L STR (1.25 L)	SmF (pellets)	N/A	1385	[49]
*Coriolus hirsutus*	10-L jar fementor (160 rpm)	SmF, free cells, semi-continuous	0.25 g/L Cu^+2^	83,830 (1st fermentation)	[50]
				80,730 (2nd fermentation)	
*Pleurotus ostreatus*	Benchtop fermenter (3 L; 200 rpm)	SmF, free cells	oil mill wastewaters	65	[51]
*Panus tigrinus*	3-L STR (2 L; 250 rpm)	SmF, free cells	oil mill wastewaters	4600	[52]
*Panus neostrigosus*	2-L STR (1.5 L; 750 rpm)	SmF (pellets)	N/A	209.96	Present study
*P. neostrigosus*	2-L STR (1.5 L; 750 rpm)	Continuous SmF (pellets)	N/A	678.49	

Note: N/A: not available; SmF: Submerge fermentation; STR: stirred tank bioreactor. *a* The volume in brackets refers to the working volume

**Table 9 jof-07-00853-t009:** Various studies on reducing sugar production from lignocellulosic biomasses.

Substrate	Pretreatment	Enzyme Used	Reducing Sugar Yield (mg/gdw)	Reference
Wheat straw/	glycerol	*Penicillium decumbens* JUA10 cellulase and β-glucosidase	900	[59]
Sweet sorghum bagasse/	dilute NaOH, autoclaving and H_2_O_2_	Celluclast 1.5 L	909	[60]
Rice husk/	alkaline peroxide-assisted wet air oxidation		210 (Glucose)	[61]
Rice hull	Alkaline peroxide	Celluclast 1.5 L and Novozyme 188	154	[30]
Corn stover, Miscanthus and wheat straw	Sodium hydroxide	Cellulase Onozuka	215, 258, and 280, respectively	[62]
Rice hull	Lime	Celluclast, Novozyme 188 and Viscostar	428	[63]
Rice straw	Steam explosion	Celluclast 1.5 L, Periconia sp. bcc2871	132	[64]
Rice husk/		*Phanerochete chrysosporium*	447	[18]
Wheat straw/	Series of white-rot fungus Euc-1	Onozuka R-10	230	[16]
Corn stover/	*Ceriporiopsis subvermispora*	Spezyme CP	565 (Glucose)	[17]
Rice hull/	*P. ostreatus*	Enzyme powder	398	[65]
Rice straw	Fungal consortium	Arrowzyme	492	[66]
Rice straw	Non-pretreatment	Celluclast 1.5 L	129	Present study
	Ligninolytic enzyme from *P. neostrigosus* I9	Celluclast 1.5 L	440	

## Data Availability

The data presented in this study are available on request from the corresponding author. The data are not publicly available due to privacy.

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
