# Peer review of "Simultaneous Biological Pretreatment and Saccharification of Rice Straw by Ligninolytic Enzymes from Panus neostrigosus I9 and Commercial Cellulase"

_jof, 2021, doi:10.3390/jof7100853_

Round 1

Reviewer 1 Report

Your work is pertinent and welcome. The demand for sugars to bioethanol production is high, and new sources or need to be study in order to improve the yield and/or decrease the costs of the process. Nevertheless, the Ms. must be improved in the organization and sequence of the “material and methods” (M&M) and “results”, both very confused sections. Also, The data presentation and the conclusions must be improved.

Important and more specific points or hints in order to improve your manuscript:

  • The title of the Ms. does not reflect the work done; considerer change it
  • Abbreviated Celsius degrees as ºC – e.g. in lines 118, 121, 126 you have several abbreviations for it
  • In point 2.2.1., tomato juice was inoculated after sterilization? It was sterilized?
  • Why put the information about modeling growth rate like this? Give the information as it is needed;
  • As I understand you performed a scaled up from 2.2.1 to 2.2.2. Some doubts: the pH was measured after sterilization – usually is before that; explain the need for sterilize the medium twice.
  • Growth mathematics, laccase production, parameters should be in a separate point, before statistical analysis
  • In line 145 – remove “population size” – in moulds growth is more correct the use of biomass.
  • Do lines 131-139 and 173-185 refer to the same - Cultivation of Panus neostrigosus I9 in continuous fermentation? The aim of this point was produce laccase. For what? Clearly explain (I suppose it was for using to pretreated rice straw)
  • Point 2.3.2 – instead of write a range of enzyme concentration, time and temperature, put the three values (10, 25, 40 U/g rice straw); time of treatment (1, 6.5, 12.0 hours) and temperature, (30, 45, 60 ºC). Also, in table 1 do not abbreviate temperature and concentration; correct the spelling of “midium”.
  • Please, confirm if all genera and species names are in italic
  • Point 2.3.3. This point is not sufficiently explained and is related to what?
  • The continuous reactor had 2.0 (point 3.1) or 1.35 (M&M) or 1.5 (Figure 1, line 352) liters?
  • As I understand Figures 1 refer to batch while fi. 2 refer to continuous growth. Correct? But the text is confuse and mix both conditions. Please, clarify.
  • How the fungi did grew in liquid medium? As puffballs, or dispersed hypha? In general, the washout (line 329) is easy to avoid when moulds are used.
  • Line 323 – correct “lagging” (lag)
  • Line 351 – which previous results? Published?
  • Total carbohydrates decreased over incubation time (Fig. 4) – if the medium flow feed the reactor continuously why the carbohydrates concentration was not constant?
  • Table 2 – some words are lowercases other uppercases; the font is different along the table. Uniform the significant numbers – here and along the Ms. – (e.g. 678.49 ±20.39 is different from 678.5 ±20.4?; a SD of 0.034 is different from 0?). Time is hours? The operation modes must be stated in M&M section
  • Table 3 – uniform the formatting
  • Line 389 – substitute “cellulase” by “cellulases” or “cellulolytic enzymes”
  • Line 391 – again, which “our previous studies”? Already published?
  • Tables 4 and 6 – it makes no sense put two significant numbers for temperature values (30.00 ºC); 30 ºC
  • Table 5 must be after line 416
  • Do not abbreviate Adj R-Squared, Pred R-Squared., Adeq Precision
  • Merge figures 5, 6 and 7, into Fig.5A, B, …; F).
  • The %delignification expression is correct? OR “)10.(“ should be “(10)“
  • Table 6 is really necessary? It only have a single lane, proper described in the text
  • The explanation given in lines 457-459 should be in M&M for a better understanding of laboratory work flow
  • In Fig. 8 substitute “non” by “non-pretreated” OR “control”. Fig. 8 and Table 7 have overlap results
  • Figure 9. – Only orange lines are present; none are blue.
  • Line 499, “cellulase enzyme” – substitute by “cellulase enzymes”. Lines 497-499; 513-514 – transferred them to M&M.
  • In discussion the results were repeated (e.g. 539-549).
  • Table 9 must appear after line 556
  • Change the phrase “In the hydrolysis phase by cellulase, the rice straw pretreated with the biological pre-treatment method was better digested”

Improved the conclusions of your work; very simple considering the amount of the results obtained.

Reviewer 2 Report

This paper described about the optimization of the productivity of Laccase from Panus neostrigosus in fermenter and the pretreatment of rice straw by the crude enzyme of P. neostrigosus. The delignification pretreatment using crude enzyme was effective for enzymatical saccharification. Experimental design is well managed and well written. However, some modification and additional information will be needed.

  1. Please check manuscript carefully. There are several mistakes, eg, line32, 37, 105, 573-575,
  2. Title is not suitable for this paper. This paper described at the pretreatment of rice straw by enzyme not fungus. This title induces the misreading the fungal mycelium was used for pretreatment and saccharification directly.
  3. Line 218. How did the author prepare crude enzyme? Purified partially? To evaluate the effect of Laccase for pretreatment, some purification steps to remove the nutrients and mycelium from crude enzyme and concentration steps should be needed. Please explain.
  4. The author focused on the Laccase only in this experiment. To eliminate the possibility of the effect of other enzyme such as peroxides, the author should be shown the enzymatical activity of manganese peroxidase, lignin peroxidase in crude enzyme.

Reviewer 3 Report

Due to its low substrate specificity lignolytic enzymes are commonly believed to be useful in bioethanol production. However even now there is need for new producers and more efficient enzymes, which are immediately applied in the mentioned above processes. In the reviewed paper authors used isolated strain P. neostrigosus producing laccase in detoxification of lignocellulose, which may in future be applied in the bioethanol production.. However in the opinion of the reviewer the paper needs some corrections listed below:

Major Compulsory Revisions:

  1. The Authors stated that neostrigosus is white rot fungus, so it should produce not only laccase, but other enzymes as well. Why only laccase was taken into consideration?
  2. The Authors should emphasize what is new in this paper comparing to previous findings.

Specific comments:

Line 115 – How the fungus was identified? Any ITS sequences were deposited to GenBank?  

Line 121 – Why tomato juice was used?

Line 124 – For shaken cultures the inoculum is often homogenized as it accelerates fungal growth. Why it wasn’t done in this way?

Line 126 – How mycelium was transferred to the fermenter jar? How medium from inoculum was removed in sterile way?

Line 133 – lbs? Please correct all units in SI.

Line 135 – 750 rpm? It seems as high rotor speed. What impeller was used?

Line 191 – What was the length of straw?

Line 194 – Crude enzyme? How it was obtained? Purified initially from mycelium?

Line 207 – hydrolyzed? Please provide reaction conditions.

Line 214 – as above, conditions….

Line 262 – Please cite original paper for laccase assay.

Fig. 1 –Why the culture was stopped in the maximum of enzyme production? What happened next? Perhaps enzyme activities were higher?

Table 3 – any citation for this composition?

Line 544 – it is hard to compare liquid cultures with SSF. What do you think?

Reviewer 4 Report

Abstract: Simultaneous Biological Pretreatment and sac- 13 clarification (SPS), correct it.

Abstract need to be revised to minimize the grammatical errors.

Line no. 37, bioreactor ]3[, correct it.

Line no. 46, The enzyme used for biological pretreatment is 46 called ligninolytic enzyme, rewrite the sentence.

Line no. 105, P. neostrigosus I9, italics

Line no. 121, and Line no. 126, 30oC and 28 oC, keep the same formatting throughout the manuscript.

Line no. 194, ground rice strawwas pretreated, correct it.

Line no. 292, Reducing sugar yield (%) formula, check spelling errors

Figure. 4. Why carbohydrate content is decreased after 48 to 108 hr and incomplete conversion? What is the substrate used here?

Table 7, why yields are low? Is it due to low enzyme dose or time?

Did you measure intracellular and extracellular laccase activity?

Authors should include the novelty of current work.

Round 2

Reviewer 1 Report

Congratulations on the significant improvements in the presentation of results and material and methods.

Some minor corrections:

Line 268 “Acid pretreatment/ “…. grams of Ground ground” – put the amount

Paragraph between lines 500-507 – change the references to fig. 5, fig 6 and fig 7 to Fig. 5(a), Fig. 5(b) and Fig. 5(c), in accord to the legend.

Lines 526-527 – remove the sentence “The delignification response of three factors in a two-stage factorial design is shown in Table 6”, since the table was deleted

The new Fig 6 seems repeated

The new Fig 7 – both the lines still orange.

Lines 559-560 – “And this problem should be solved in the next experiment” – considerer change to “And this problem should be solved in future experiments”
